*Review Article*

# Graded versus ON/OFF control in quantitative gene expression and epigenetic memory

Caroline Dean [1]✉ & Martin Howard [2]✉

## Abstract

**How can expression of a specific gene be quantitatively regulated? In this review, we discuss two possible modalities. In one, the level of mRNA generated from each gene copy can be smoothly varied giving graded analogue control. In a second, some gene copies generate high mRNA levels whilst others generate very low levels, giving ON/OFF digital control, with the fraction of copies with high or low expression being regulated. We focus on why in different contexts one modality would be preferred over the other, how these two modalities can be generated through transcriptional regulation, and discuss whether ON/OFF control is particularly linked to epigenetic memory. We argue that digital control arises for memory mediated by trans-factor feedback loops and histone modifications, but not necessarily for DNA methylation. We also examine how these expression modes can be established at one specific target, Arabidopsis *FLOWERING LOCUS C (FLC)*. Graded expression and switching to ON/OFF control occur during early development and during long-term cold exposure and both are key to *FLC* regulation.**

**Keywords** Quantitative Gene Expression; Epigenetic Memory; Polycomb System; ON/OFF Versus Graded Control; Mathematical Modelling
**Subject Categories** Chromatin, Transcription & Genomics; Computational Biology; Plant Biology

## Introduction

Our understanding of transcriptional regulation has increased enormously in recent years (see Buratowski and Arndt, 2025) and references therein). Assembly of the pre-initiation complex (including RNA polymerase II (Pol II), TFIID and Mediator) (Haberle and Stark, 2018) is followed by many other steps that regulate transcription. After transcription of a few nucleotides, the Pol II pauses, in a process tightly regulated by factors including NELF (Adelman and Lis, 2012), and either terminates or continues into productive elongation (Jonkers and Lis, 2015; Muniz et al, 2021). Further Pol II elongation control and co-transcriptional splicing occur, but always with the possibility of premature

termination (Carrocci and Neugebauer, 2024; Bentley, 2025). All of these processes occur within a local chromatin environment where DNA methylation and the histone modification landscape play important regulatory roles (Szczurek et al, 2024; Wang et al, 2023; Chan et al, 2022). Finally, release of the full-length transcript from the chromatin is linked to export into the cytoplasm, where degradation of the transcript provides an additional level of regulation.

These varied steps in transcriptional control have been characterised quantitatively utilising a variety of biophysical, sequencing, imaging and mathematical approaches (Jeronimo et al, 2021; Chatsirisupachai et al, 2025; Shao et al, 2022; Uzun et al, 2021). Similarly, processes upstream of initiation, including the dynamics of transcription factors and enhancer/promoter interactions, as well as the dynamics of downstream translation, have also been quantitatively investigated (Izeddin et al, 2014; Barshad et al, 2023; Li et al, 2019; Doughty et al, 2024; Mahendrawada et al, 2025). In this review, we will take a slightly different, albeit complementary, perspective on the regulation of gene expression. We first zoom out from the detailed individual steps of transcription and consider the fundamentally different types of quantitative gene expression control. We argue that two modes are possible, graded versus ON/OFF control, discuss why different modalities would be used, and then describe a specific gene (Arabidopsis *FLC*) exemplifying the different modalities, examining how these modes can be linked together. Our intention is to stimulate discussion of the different modes of quantitative gene expression control to progress our fundamental understanding of both transcription and epigenetic memory, including the transcriptional events that can stimulate epigenetic switches.

## Graded versus ON/OFF modes of quantitative gene expression control

If the quantitative expression levels (here defined as the level of mRNA) at a particular gene are important, how are the correct levels achieved? At a conceptual level, we can distinguish two modes of quantitative gene expression regulation (Fig. 1A). The first is where the expression can be smoothly varied up or down like a dimmer (Fig. 1A right). We will also refer to this graded mode of regulation as analogue, where above a lower background limit, and below an upper limit of maximal expression, any level of expression is possible. The second mode is where the expression from each

[1]Department of Cell and Developmental Biology, John Innes Centre, Norwich Research Park, Norwich NR4 7UH, UK. [2]Department of Computational and Systems Biology, John Innes Centre, Norwich Research Park, Norwich NR4 7UH, UK. ✉E-mail: caroline.dean@jic.ac.uk; martin.howard@jic.ac.uk

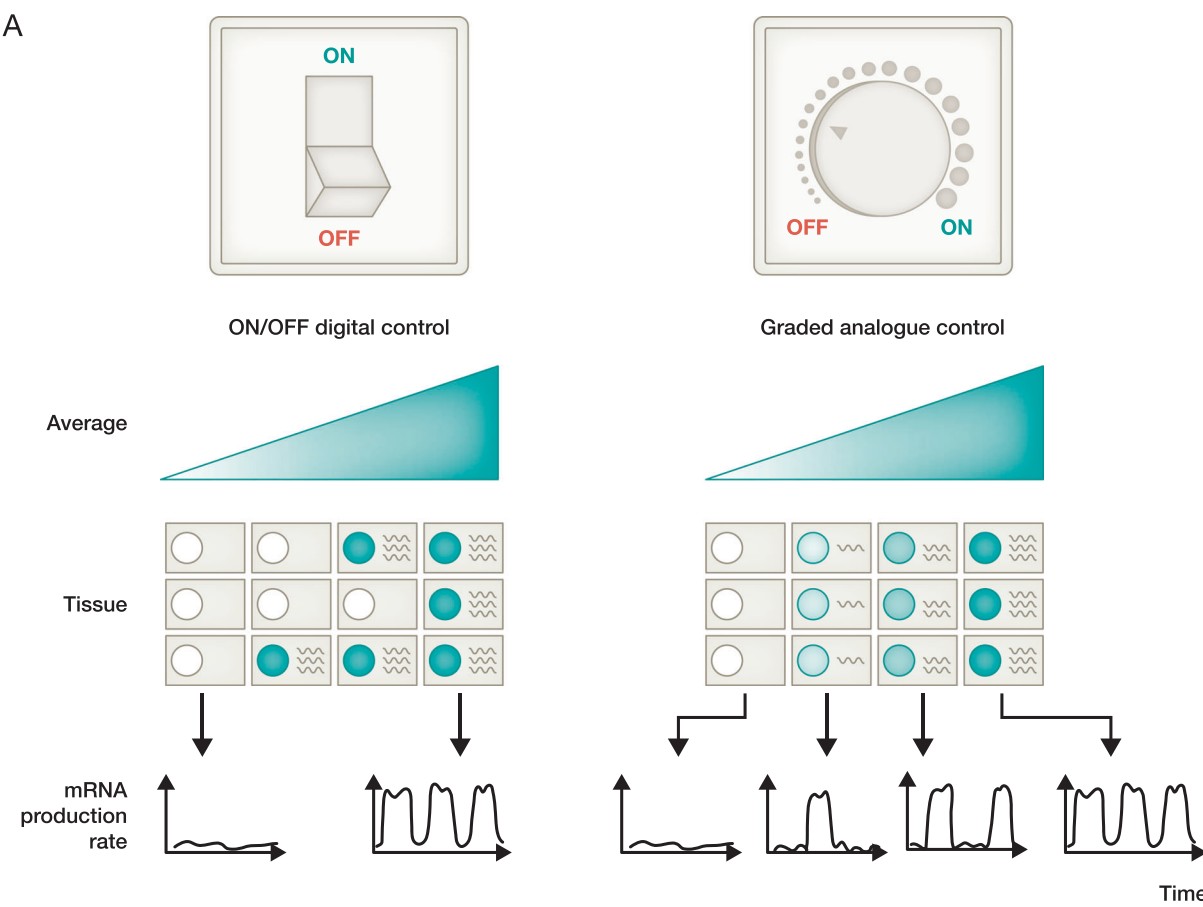

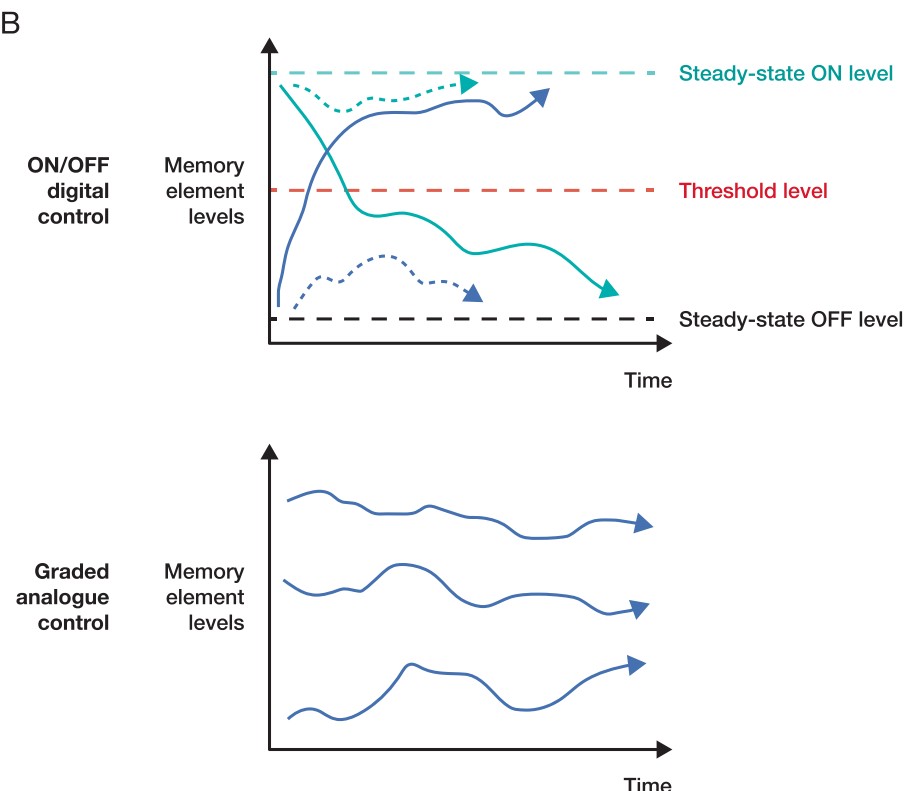

**Figure 1. Comparing ON/OFF (digital) and graded (analogue) modes for control of quantitative gene expression.**

(A) Left: ON/OFF regulation, where quantitative control comes from the fraction of loci that are ON or OFF. Right: graded regulation, where the quantitative expression of each locus can be smoothly varied. Bottom: schematics showing transcriptional output as a function of time, illustrating that both ON/OFF and graded modalities are compatible with bursty gene expression. (B) Top: ON/OFF memory element, with steady-state ON (green dashed line) and OFF (black dashed line) levels, separated by a threshold level (red dashed line) which must be passed for transitions from one to the other. Small fluctuations fail to pass the threshold (dotted curves with arrows). The dynamics then pushes the system back towards the previous steady-state. Only large fluctuations can pass the threshold and cause a switch from ON to OFF or vice versa (full curves with arrows). Bottom: Graded memory element. Even small fluctuations away from the starting value are not corrected. This will cause analogue memory element levels to drift stochastically away from their starting values, eventually eroding the memory (black curves with arrows). DNA CG methylation may avoid this fate, at least for sufficiently short times, due to the very low magnitude of the fluctuations.

gene copy is restricted to high or very low values only, with intermediate levels not occurring (Fig. 1A left). We will refer to this mode of regulation as ON/OFF or digital. For simplicity, we will first discuss the two modalities as being entirely separate; of course, it is perfectly possible that they can be combined. For example, in an otherwise digital modality, the level of the ON expression state could be smoothly modulated up or down in an analogue way.

If we consider whether a given gene is being actively transcribed or not at any given instant of time, transcription is fundamentally ON or OFF (either a gene is being actively transcribed by one or more elongating Pol IIs, or it is not). However, it is important to make a distinction between such an instantaneous snapshot of gene activity (transcription) versus mRNA levels (expression), with the latter being essentially a time-average of transcription over the lifetime of the resulting mRNA. It is for the mRNA expression level, a more coarse-grained quantity with a longer timescale, that the graded versus ON/OFF classification becomes relevant.

In the analogue modality, each gene copy can express at any particular level and hence an individual cell, or a collection of cells, can, in principle, generate the full range of quantitative regulation without the need for any further regulatory concepts (Fig. 1A right). Generating such output may nevertheless require sophisticated regulation to limit cell-to-cell expression variability, for example. However, for the ON/OFF mode, clearly an individual diploid cell has only three levels of expression (OFF/OFF, ON/OFF, ON/ON), if each copy of the gene can be independently regulated, or only two (OFF/OFF, ON/ON), if both copies share the same regulation. For a collection of cells, an overall smoothly varying quantitative expression level can still occur through varying the fraction of gene copies that are ON or OFF. This overall method of regulation, where the fraction of ON or OFF gene copies is varied, has sometimes been referred to as 'fractional control' (Fig. 1A left) (Bintu et al, 2016; Mukund and Bintu, 2022). During the G2 phase of the cell cycle, the gene copy number doubles and hence more complex ON/OFF patterns are possible, but these will rapidly resolve to the above cases during the next cell cycle. Only for highly endoreduplicated cells, with many chromosome copies, will it be possible for digital regulation at each gene copy to approach analogue control at the single-cell level.

Transcriptional modes have also been divided into bursty versus non-bursty (Tunnacliffe and Chubb, 2020). For bursty transcription, transcripts from a gene copy are produced in discrete bursts with quiescent gaps in time with little or no transcript production, due, for example, to reduced initiation. Alternatively, in non-bursty transcription, transcripts can simply be produced with an overall constant probability per unit time, without any temporal clustering of transcription as in the bursty case. We emphasise that both

bursty and non-bursty transcription is completely compatible with our distinction into graded versus ON/OFF control for expression. For non-bursty, the overall probability that a complete transcript is produced is either high or low for ON/OFF or can be smoothly varied for the graded mode. For bursty transcription, the frequency of transcriptional bursts can, for example, be varied (Weinberger et al, 2012): again, high or low for ON/OFF versus smoothly varying for the graded modality (Fig. 1A, bottom). For cases where the mRNA lifetime is much greater than the timescales associated with transcript production (e.g. the timescales of burst duration or between bursts), the two modalities of analogue and digital emerge at the level of gene expression.

The key distinction between graded and ON/OFF quantitative gene expression regulation then raises several questions:

- What determines whether graded or ON/OFF control is used for a particular gene?
- In terms of the detailed individual steps of transcription, what mechanisms can ensure graded or ON/OFF expression?
- How do these two fundamentally different modes interact with each other, particularly at genes that may transition from one mode of regulation to the other?

## Why would graded or ON/OFF control be used at a particular target?

ON/OFF control at the level of an individual gene copy only allows for a few discrete levels of expression, which could be a disadvantage. However, the key property for the ON/OFF digital mode that can eclipse this issue is its enhanced robustness as compared to graded control. Biological systems need to be robust against fluctuations, both internal within their regulatory network (intrinsic biochemical noise and extrinsic cell-to-cell variation) as well as external from the environment (Barkai and Shilo, 2007). Conceptually, ON/OFF systems demonstrate enhanced robustness regardless of their molecular implementation: for well-separated ON and OFF levels, only very large fluctuations will be able to flip the state of the system from ON to OFF or vice versa by crossing the stability threshold that separates them (Fig. 1B, top). Graded, analogue systems, on the other hand, will display far less robustness as their fluctuations cannot easily be corrected, see below (Fig. 1B, bottom).

This greater responsiveness may be advantageous when the system needs a range of responses to variable inputs. However, one scenario where such sensitivity will be problematic is in systems that must exhibit memory, i.e. in epigenetic regulation. Here, a

gene must 'remember' its expression level even when the originating signals that first set that level have disappeared (Ptashne, 2014). In this case, a graded, analogue system, regardless of its molecular implementation may perform poorly, as any fluctuation-driven drift in its expression level cannot be corrected due to the absence of the regulatory input signal (Fig. 1B, bottom). For this fundamental reason, an ON/OFF digital expression mode is often likely to be required for epigenetic memory. As we will see below, this conclusion applies to trans-factor- and histone modification-mediated epigenetic systems, particularly the Polycomb system, a conclusion that has now been verified experimentally for several cases (Angel et al, 2011; Berry et al, 2015; Ng et al, 2018; Movilla Miangolarra et al, 2024; Saxton and Rine, 2019; Klosin et al, 2017; Bintu et al, 2016; Menon et al, 2021). Overall, we propose that graded, analogue systems will be the most common expression modality, whereas an ON/OFF mode will often be favoured by systems requiring long-term epigenetic memory over multiple cell cycles. However, ON/OFF control may also be favoured in other specific contexts, e.g. activation of the transcription factor NFκB by tumour-necrosis factor (TNF)-α (Tay et al, 2010), transcriptional activity in response to heat shock in *Arabidopsis* (Alamos et al, 2021) and in the p53 pathway (Hafner et al, 2020).

## Generating graded gene expression

Traditionally, gene expression is considered to be limited by the binding of transcription factors that recruit the necessary transcriptional machinery to initiate transcription. An excellent example where a varying concentration generates graded expression is the transcription factor Msn2 for stress-responsive genes in budding yeast (Stewart-Ornstein et al, 2013). This relationship was shown to be generated through noncooperative low-affinity interactions between Msn2 and its cognate binding sites (Stewart-Ornstein et al, 2013). However, the transcription factor paradigm can now be expanded to include the many steps of transcriptional regulation discussed above (Diao et al, 2025; Bentley, 2025). Furthermore, transcription and RNA processing are no longer considered as sequential processes but are rather intimately coupled co-transcriptionally. Transcription delivers chromatin modifications, which influence RNA processing, which in turn delivers further chromatin modifications that affect subsequent Pol II processivity and transcriptional output (Chalamcharla et al, 2015; Vasiljeva et al, 2008; Menon et al, 2024; Chan et al, 2022). A case of graded expression utilising a two-way relationship between transcription and RNA processing can be found at *FLC* and will be discussed in more detail below (Menon et al, 2024).

In many cases, even for graded expression, the various feedbacks can make predicting quantitative expression levels very complex. Indeed, mechanistic understanding often necessitates a close combination of theoretical modelling and experiments (Stewart-Ornstein et al, 2013; Menon et al, 2024), an issue we will return to below. This was also elegantly demonstrated in the dissection of the regulation of budding yeast GAL1 transcription. In this case, sense transcript production and processing are reduced (and transcript stability is increased) by high rates of antisense transcription (Brown et al, 2018).

## Generating ON/OFF control through two different mechanisms

As discussed above, ON/OFF regulation is often likely to be a prerequisite for epigenetic memory due to its inherent enhanced robustness. Here, we discuss two conceptually different mechanisms that allow ON/OFF gene expression control.

One ON/OFF system is a trans-factor-based mechanism, where the gene of interest is able to directly or indirectly strongly auto-activate itself, given sufficiently high levels of expression (Veening et al, 2008). Once expression, and the resulting protein concentration level, have passed a critical threshold, this will drive the gene to high expression levels (ON). For low concentration levels, however, the feedback is insufficiently strong, and a low expression state persists (OFF). In this case, the memory is held in the concentrations of the resulting protein, which can then be inherited by daughter cells at cell division, recapitulating the memory state (Veening et al, 2008). Such memory states can be very stable, as, provided the activating factor concentration remains high enough to sustain the feedback loop, the memory will persist, allowing the system to buffer large concentration fluctuations. Such systems have been frequently discussed in the literature, with the Lambda repressor being the classic example (Ptashne, 2014). In that case, the Lambda repressor encodes an auto-activating positive feedback loop, maintaining lysogeny, while stably repressing transcription of lytic genes. In this way, the lysogenic state is stable and only rarely flips to the lytic state in the absence of external signals (Ptashne, 2014).

A second ON/OFF system is a cis-based memory system, which can occur where the memory is encoded in the local chromatin environment in the form of histone modifications. Histone modification memory partly depends on the read/write activity of the enzymes that add the histone modifications, where the enzymes can bind to already existing marks and then add more of the same mark nearby in a positive feedback loop. For sufficiently high local levels of these histone modifications, the read-write dynamics generate strong positive feedback, enough to maintain these high levels despite the presence of fluctuations. Low levels of the modifications do not generate sufficient feedback, maintaining low modification levels and permitting an opposite expression state. The complexities of these feedback loops can make the resulting dynamics hard to intuit. For this reason, mathematical modelling has again been invaluable in dissecting the underlying mechanisms (Dodd et al, 2007; Angel et al, 2011; Movilla Miangolarra et al, 2024; Briffa et al, 2024; Richards et al, 2012; Berry et al, 2017; Sneppen and Dodd, 2012; Michieletto et al, 2018; Jost and Vaillant, 2018; Mutzel et al, 2019; Lövkvist and Howard, 2021; Owen et al, 2023).

Until recently, it was controversial whether such cis-based systems could genuinely hold memory or whether the chromatin modifications were merely the consequences of persistent expression states where the memory was held in trans-factor feedback loops (like for the Lambda repressor). Recent cellular-resolution experiments have directly addressed this issue, often involving labelling of proteins from the two gene copies with distinguishable fluorophores. Such experiments have demonstrated that an individual gene copy in a diploid organism can maintain an independent expression state heritable through DNA replication and cell division, not necessarily shared by the other copy (Berry

et al, 2015; Ng et al, 2018; Holoch et al, 2021; Klosin et al, 2017). A trans-factor memory system would generate the same memory state at both diploid gene copies, due to the memory effectively being stored in a shared cellular pool. Hence, these experiments have provided clear evidence in favour of cis memory.

Why have many genes evolved histone modification-based systems rather than trans-feedback loops to hold memory? Trans-factor memory, such as for the Lambda repressor, is clearly evolutionarily ancient and a viable solution to the problem. However, the appropriate positive feedback loops will need to evolve for each target in question. Potentially, therefore, cis-based epigenetic memory systems may predominate as these mechanisms can be employed as an 'off the shelf' feedback system that can be reused to control any gene.

Histone modification-based memory systems must provide reliable memory despite the very large fluctuations they are subject to, particularly at DNA replication. During this phase of the cell cycle, approximately half of the histone modifications will be lost due to the distribution of parental nucleosomes between the two daughter DNA strands. We argue that the need to recover from such a large perturbation necessitates an ON/OFF digital system. Provided sufficient marks survive replication, the strong read-write action of histone modifiers would fill in the missing marks. If histone modifications encoding memory were a graded analogue system, then following DNA replication, the system would have to increase modification levels twofold to recover the initial levels. In fact, the requirement may be even more complex, as a halving of histone modification levels is believed to only occur on average (Annunziato, 2005). Due to the stochastic inheritance of nucleosomes, the fraction of modifications surviving on a given daughter strand may be either greater or smaller than a half (Annunziato, 2005). An analogue histone modification memory system would have to register this survival information and then incorporate it into the post-replication recovery to reacquire the pre-replication graded level of histone modifications. This modality would therefore inevitably require a very complex regulatory system. For these reasons, analogue memory for histone modifications is unlikely, with a digital ON/OFF memory system being much more robust and consistent with the read-write feedback observed experimentally (see below).

## ON/OFF control in the Polycomb memory system

A particularly conserved example of histone-based ON/OFF control, and one which has been the subject of intensive study, is the Polycomb system and its cognate histone modifications H3K27me3 and H2AUb (Lewis and Mislove, 1947; Lewis, 1978). Distinct complexes of Polycomb Group (PcG) proteins, known as Polycomb Repressive Complex 2 (PRC2) and Polycomb Repressive Complex 1 (PRC1) were shown to impart a cis-acting transcriptional silencing to the local chromatin (Schuettengruber et al, 2017; Xiao et al, 2017; Simon and Kingston, 2013; Blackledge and Klose, 2021; Bienz and Müller, 1995; Francis et al, 2001). The conservation of Polycomb silencing mechanisms across flies, mammals, and plants has inspired questions about how epigenetic states are established and how they integrate environmental and developmental signals.

Polycomb targets exhibiting memory can exist in ON or OFF states: epigenetically ON, marked by activating modifications such as H3K4me3, or OFF, marked by silencing modifications such as H3K27me3 and H2AUb. Key to maintenance of the OFF state is the read-write action of PRC2, where the complex can 'read' existing H3K27me3 marks, and then undergo allosteric activation to 'write' more of the mark nearby in a positive feedback loop (Uckelmann and Davidovich, 2021). Related feedback loops also exist for H2AUb via PRC1 (López et al, 2024; Uckelmann and Davidovich, 2021), as well as for PRC accessory proteins that can polymerise (Fiedler et al, 2022; Schulten et al, 2025; Payne-Dwyer et al, 2025; Frey et al, 2016). Crosstalk between H2AUb and H3K27me3 has also been demonstrated via PRC2 complexes containing the H2AUb-binding JARID2 subunit (Shi et al, 2024). These multiple feedback mechanisms allow for the diluted levels of silencing histone modifications following DNA replication to be restored during the subsequent cell cycle (Reverón-Gómez et al, 2018). Transcription and Polycomb silencing also mutually inhibit each other (Berry et al, 2017) in a tug-of-war mechanism, where the act of transcription disrupts the silencing histone modifications, through, for example, the co-transcriptional activity of demethylases. Transcription is itself stimulated by the action of Trithorax Group (TrxG) proteins. These proteins were characterised genetically in Drosophila (Schuettengruber et al, 2011) and then identified as a diverse group of chromatin-modifying proteins that activate gene expression. They deliver histone modifications that promote transcription (Klymenko and Müller, 2004; Kingston and Tamkun, 2014; Morgan and Shilatifard, 2020), though their function may be more accurately thought of as anti-repressors, rather than activators. Together, through these mechanisms, the active ON state can also be stabilised.

Importantly, not every Polycomb target exhibits digital ON/OFF memory. Even in more differentiated cell types, such as mouse embryonic fibroblasts and neural progenitor cells, only a fraction of Polycomb targets appear to possess this property, at least as can be determined from experiments with transient inhibition of PRC2 (Holoch et al, 2021). Hence, the digital epigenetic memory systems discussed here will be dispersed among a wider set of targets without these features. Indeed, different strengths of feedback mechanisms will affect the timescales of memory maintenance, and these may vary at different gene targets and across cell types. Reactivation dynamics of the silenced state with H3K27me3 dilution/loss has been studied at the X chromosome (Kohlmaier et al, 2004) and at mammalian developmental genes (Jadhav et al, 2020). The contrasting stabilities of Polycomb-mediated memory states is an important area for future dissection.

How Polycomb silences at the level of the detailed steps of transcription has been a live question for some time. The original idea of active Polycomb-induced compaction of the chromatin blocking access of transcriptional activators to DNA elements is inconsistent with many recent experiments (Uckelmann and Davidovich, 2024; King et al, 2018). Drosophila PcG silencing (dependent on the PRC1 factors *Pc*, *Pcl*, *Psc*, *Scm*) was found to directly affect the activity of the transcriptional machinery at the promoter (Dellino et al, 2004). In mouse Embryonic Stem Cells, PRC1 promotes a stable OFF state by counteracting the binding of factors that enable early Pol II pre-initiation complex formation (Szczurek et al, 2024), although PcG repression is mostly associated with genes capable of only low levels of expression (Berrozpe et al,

2017). This finding accords with transcription being antagonistic to Polycomb silencing (Berry et al, 2017) and is a result shared with yeast heterochromatin (Wu et al, 2024). PcG repression has also been associated with paused Pol II at promoters, and thus with altered transcriptional elongation (Enderle et al, 2011), as well as with inhibition of Pol II release from the transcription start site (Zhang et al, 2020).

A vital aspect of Polycomb dynamics is how systems switch into the silenced state. In general, such a switch to an OFF state occurs via distinct phases, with an initial requirement for recruitment of factors to a nucleation site (Laprell et al, 2017) or CpG island (Oksuz et al, 2018; Laugesen et al, 2019), followed by histone modification spreading in a process requiring PcG activity and an active cell cycle (Yang et al, 2017; Oksuz et al, 2018; Jiang and Berger, 2017). The best example of an environmentally mediated histone modification switch of an endogenous gene is Polycomb regulation at *FLC* in Arabidopsis. Long-term epigenetic memory at this target is well established (Yang et al, 2017; Gendall et al, 2001; Bastow et al, 2004; De Lucia et al, 2008; Qüesta et al, 2020) and we will discuss its behaviour in much more detail below.

## The case of DNA methylation may be more complex

Another class of epigenetic memory systems are those mediated by DNA CG methylation, which in many cases result in gene silencing (Greenberg and Bourc'his, 2019), although its role in plants is more intricate (Zhang et al, 2018). Interestingly, DNA methylation systems may not necessarily be subject to such a strong ON/OFF constraint when it comes to maintenance of the memory state. For DNA CG methylation, maintenance can occur at the level of each individual CG site, where hemi-methylated CGs are recognised and remethylated during the subsequent cell cycle. This is in contrast to the histone modification-based systems, where it is only the collective state of the histone modifications that stores memory. This distinction is clearest for an isolated individual histone modification, which might be stochastically entirely lost at DNA replication, whereas an isolated mCG becomes hemi-methylated at replication and therefore in principle the mark can survive if remethylation is sufficiently strong. This difference is key to understanding how DNA methylation and histone modification memory states may differ. Strong maintenance of individual methylated CGs allows any graded DNA methylation level to potentially be maintained at a locus with only very slow drift caused by imperfect remethylation and/or spontaneous/cooperative methylation addition. Such a lack of digital ON/OFF memory states has already been demonstrated for CG methylation in plants (Briffa et al, 2023), and such states have been called 'analogue memory' (Palacios et al, 2025). Hence, DNA methylation memory can escape the ON/OFF constraint in cases where DNA methylation is propagated with very high fidelity (low noise). However, this issue is not entirely resolved, as other studies, including computational work based on human epigenomes, have emphasised ON/OFF methylation states (Lövkvist et al, 2016). In these cases, where remethylation of individual hemi-methylated sites may be highly imperfect, sustaining highly methylated memory states will require cooperative feedback conceptually similar to read-write histone modification feedback. Such digital output would be appropriate for

the mono-allelic expression seen in imprinting, for example (SanMiguel and Bartolomei, 2018). Overall, methylation may fall into both graded and ON/OFF modalities depending on the specific organism and locus in question.

## Arabidopsis *FLC*—a case study exemplifying the different expression modalities and how they link together

So far in this review, we have conceptually examined different modalities for quantitative gene expression, arguing that an ON/OFF mode will often be required for systems exhibiting long-term epigenetic memory. In the next part of the review, we use Arabidopsis *FLC* as an example to examine how the graded and ON/OFF expression modes are established, how they interchange and how that generates a stable epigenetic memory of silencing (Yang et al, 2017; Gendall et al, 2001; Bastow et al, 2004; De Lucia et al, 2008; Qüesta et al, 2020).

Arabidopsis *FLC* encodes a MADS-box protein that represses the transcription of the genes required for the meristem to switch to a floral state (Michaels and Amasino, 1999; Sheldon et al, 2000). It has become an excellent system to analyse expression modalities and epigenetic switches because quantitative variation in expression and cold-induced epigenetic silencing are central to reproductive timing and adaptation in a range of plant species. Although playing a major role in the timing of reproductive development, FLC also regulates many targets unrelated to flowering. Chromatin-IP analysis identified >500 genes where FLC is enriched. However, the *FLC* locus is not one of those, arguing against an autoregulatory function (Deng et al, 2011).

*FLC* expression is regulated in a graded manner in different genotypes, but switches to a digital OFF state both during early development (Antoniou-Kourounioti et al, 2023), and after progressive exposure to prolonged cold (Yang et al, 2017; Nielsen et al, 2024; Angel et al, 2011, 2015) (Fig. 2). The OFF silenced state after cold is epigenetically maintained through many cell divisions in the subsequent warm and is one of the most stable Polycomb-silenced states so far characterised. We review our understanding of this regulation below.

## Graded expression variation between genotypes

Quantitative, graded *FLC* expression regulation is a major contributor to reproductive strategy in Arabidopsis and its Brassica relatives (Zhai et al, 2024; Antoniou-Kourounioti et al, 2023). A high expression of *FLC* established during embryo formation (Tao et al, 2019) necessitates an overwintering strategy and a requirement for long-term cold before flowering, and thus only one generation per year. Constitutively low *FLC* expression underpins rapid cycling, with multiple generations possible per year, a strategy also typical of most Arabidopsis accessions used in labs. In natural conditions, a graded range of *FLC* expression levels between these extremes underpins adaptation of different Arabidopsis accessions to different climates (Shindo et al, 2005, 2006; Hepworth et al, 2020) and, where environmental conditions allow, very low expression determines a rapid-cycling reproductive strategy.

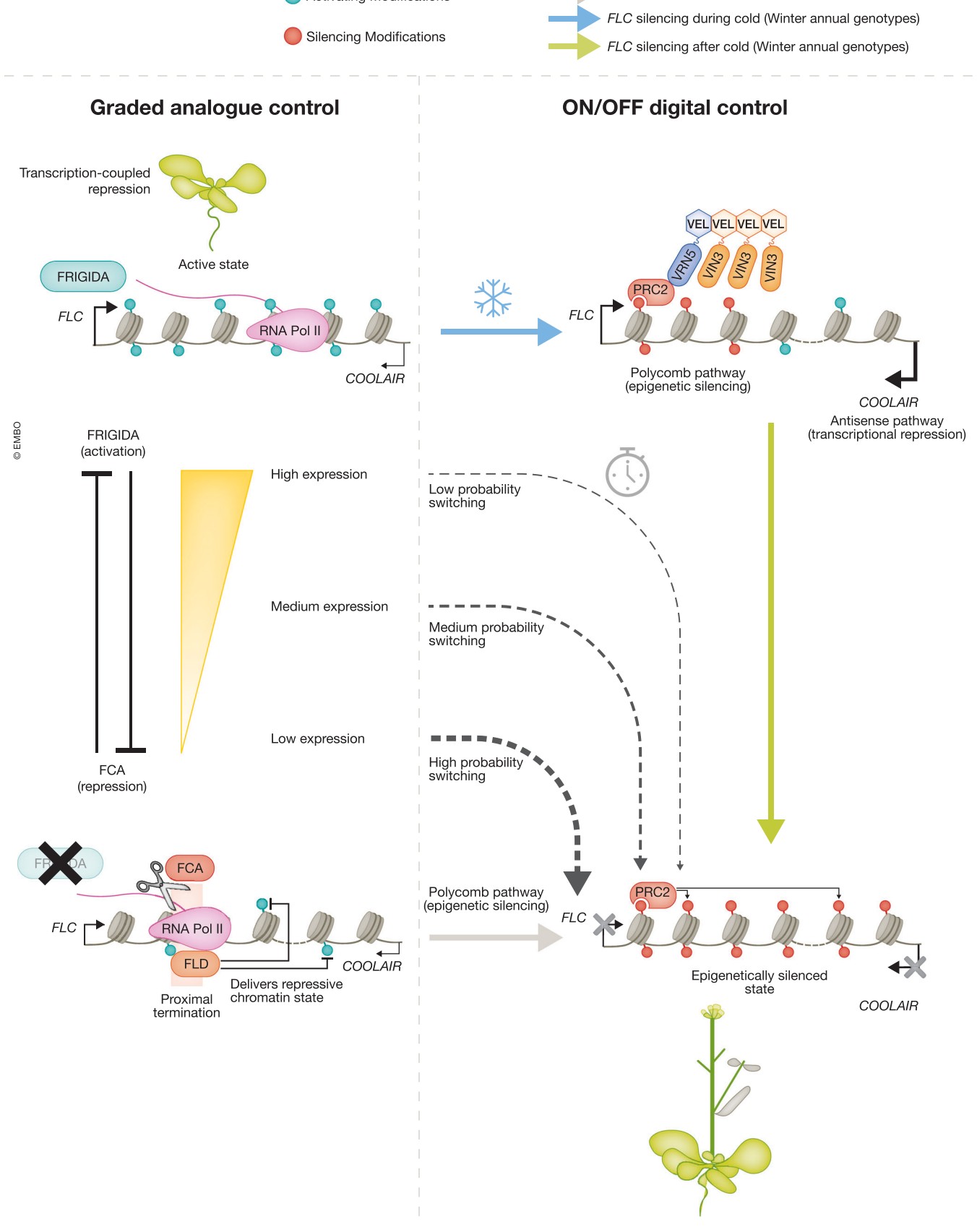

◀ **Figure 2. Schematic of chromatin regulation at *FLOWERING LOCUS C* (*FLC*).**

Left: Graded analogue control mediated by the antagonism between FRI (activation) and FCA (co-transcriptional repression mediated via FLD). Right: ON/OFF digital control mediated by the Polycomb epigenetic system. Epigenetic silencing can be triggered by prolonged cold temperatures (top right) utilising VEL protein (including VIN3 and VRN5) head-to-tail polymerisation, and H3K27me3 nucleation. H3K27me3 spreading in the subsequent warm (bottom right) contributes to stable maintenance of epigenetic memory. Switching to the OFF state can also occur in the warm, but with a variable time delay controlled by the analogue level of expression (centre). Note that non-epigenetic, analogue downregulation in the early cold has been omitted for clarity.

The *FLC* expression state is regulated through a bidirectional interplay between chromatin and transcription. The high expression state, with >20-fold higher transcription initiation, and ~10-fold increase in Pol II elongation rates (Wu et al, 2016), is mediated through activity of the transcriptional activator FRIGIDA (FRI), which determines much of the quantitative variation of *FLC* expression (Johanson et al, 2000). FRI is a lysine-rich coiled coil protein that interacts with factors suggesting it operates in a functionally equivalent manner to the mammalian RNA Pol II elongation factor ELL (11–19 lysine-rich leukaemia) (Geraldo et al, 2009). FRI-mediated transcription, as well as the action of the THO/TREX complex (Xu et al, 2021), delivers active chromatin modifications (e.g. H3K4me3) and highly processive transcription. FRIGIDA can therefore be considered an anti-terminator (Schon et al, 2021), with rapidly processed and exported transcripts (Zhu et al, 2021; Li et al, 2018).

Antagonising FRI and repressing *FLC* is the function of a series of activities initially described as the autonomous flowering pathway, including the RNA-binding protein (ELAV-like) FCA (Wu et al, 2020). High levels of expression with activating modifications can also occur in mutants of this pathway even without the activity of FRI (Wu et al, 2016). For the FCA-mediated pathway, the genetically identified activities all involve conserved 3' RNA processing factors (Fang et al, 2020) and chromatin modifiers, such as the histone demethylase LSD1-like, FLD (Liu et al, 2007). These factors promote proximal termination of transcripts from both sense and antisense *FLC* strands (the latter called *COOLAIR*). Proximal termination leads to recruitment of the histone H3K4 demethylase FLD and changed chromatin during post-cleavage transcription (Mateo-Bonmatí et al, 2024). The demethylated H3K4 chromatin promotes further proximal termination, potentially through reduced processivity of subsequent Pol II transcription, hence forming a feedback mechanism which required a combination of theoretical modelling and experiments to fully dissect (Menon et al, 2024). ARGONAUTE1 (AGO1) was also found to genetically function in this early termination mechanism, physically interacting with the spliceosome NineTeen Complex (NTC) and related proteins (NTR), as well as with the THO/TREX complex (Xu et al, 2021). Overall, FRI and FCA function antagonistically (anti-termination versus termination) in a transcription-coupled repression mechanism affecting Pol II transcription initiation, processivity and elongation, where a balance between FCA and FRI function quantitatively regulates *FLC* expression in a graded manner at each gene copy (Fig. 2, left) (Menon et al, 2024).

Evolution has clearly selected these complex co-transcriptional mechanisms as opposed to conceptually simpler transcription factor binding variation for quantitative regulation of *FLC*. One intriguing possibility is that co-transcriptional regulation has the ability to provide a wide range of graded expression levels that give an evolutionary advantage. One could imagine that the sense-antisense circuitry of *FLC*, both affected by the various co-

transcriptional factors, but both feeding back on each other through chromatin modifications affecting both strands, provides a way for small changes in factor functionality or local SNPs distributed throughout the locus to have relatively large transcriptional effects.

A transition from graded expression to Polycomb ON/OFF regulation occurs both during early development (Fig. 2, centre) and also during exposure of the plant to prolonged cold (Fig. 2, right), the latter process being known as vernalisation. Genetic analysis has shown that different regulators are involved in these two transitions, but conceptually they are very similar: both are to a digital OFF state, and both are driven by local chromatin (i.e. cis-regulated). We next discuss these transitions.

## Developmental transition from graded to ON/OFF modality

Transcription and Polycomb silencing mutually inhibit each other (Berry et al, 2017) in a tug-of-war mechanism. Hence, if transcription is sufficiently reduced at designated targets, Polycomb proteins sampling permissive chromatin sites (Klose et al, 2013) can gain a foothold and enable a stochastic switch to a digital Polycomb-silenced OFF state (Berry et al, 2017). In this way, graded and digital regulation are sequentially combined at *FLC* (Antoniou-Kourounioti et al, 2023; Menon et al, 2024). The timescale for switching between these states is controlled by the strength of the initial graded expression. For higher expression, switching to OFF is slow (low probability), while for lower expression, switching is fast (high probability) (Fig. 2, centre). The latter situation occurs in the embryo in the Col-0 genotype, which is predominantly used in labs internationally, resulting in constitutive silencing throughout development and a rapid-cycling phenotype (Schon et al, 2021). Intermediate levels of expression, generated through the use of intermediate alleles (e.g. *fca-3* (Antoniou-Kourounioti et al, 2023)), delay the switching to OFF so that it can be observed at the young seedling stage. The digital switch to Polycomb silencing is currently modelled as a stochastic event, which fits existing data (Antoniou-Kourounioti et al, 2023; Menon et al, 2024). However, as additional information becomes available, it remains a possibility that the timing of the switch may be partly controlled by other processes, including, for example, the cell cycle, which may influence diverse aspects of H3K27me3 accumulation.

## Vernalisation: a cold-induced transition from graded to ON/OFF modality with high epigenetic stability

Switching to a Polycomb-silenced OFF state also occurs during prolonged cold. Cold exposure initially represses *FLC* expression in

a graded manner in a process involving co-transcriptional splicing changes combined with reduced productive transcription initiation (Maple et al, 2025) (for clarity, this phase is not shown in Fig. 2). A lengthening duration of long-term cold also slowly induces expression of a Polycomb accessory protein VIN3. This induction relies on an indirect temperature-sensing mechanism, where NTL8, a regulator of VIN3 that is particularly stable, slowly accumulates in a graded way as dilution via cell division slows in the cold (Zhao et al, 2020).

VIN3 together with its constitutively expressed homologue VRN5 (also referred to as VIL1) form a complex with PRC2 that then nucleates H3K37me3 silencing over a small number of nucleosomes (3–4) around the first exon-intron boundary of FLC (Fig. 2, top right). Efficient nucleation of high H3K27me3 levels requires DNA sequence-specific binding of the transcriptional repressor VAL1 (Qüesta et al, 2016), similar to the necessity for Polycomb Response Elements (PREs) in Drosophila. Excising these Drosophila PRE regions leads to collapse of the histone modification states (Laprell et al, 2017; Coleman and Struhl, 2017). However, in the case of FLC, it has recently been shown that the sequence specificity modulates transcriptional activity (the graded modality), antagonising PcG activity, rather than directly affecting PcG nucleation (the digital ON/OFF modality), as in the classic PRE model (Franco-Echevarría et al, 2023).

Single-cell fluorescent monitoring experiments (Yang et al, 2017), in conditions where plants grow minimally, show that the nucleated state is established independently at each gene copy and confers metastable epigenetic silencing. The overall nucleation process occurs very slowly, over 4–10 weeks, depending on the Arabidopsis accession. The slow, cis nature of the collective switching behaviour is assured by the low probability of the nucleation process, causing each gene copy to behave effectively independently over a timescale of weeks. Once a sufficient number of nucleating proteins associate with FLC, strong positive read-write feedbacks then switch the system to a nucleated OFF state at each gene copy. The digital switch is thought to be stochastic, with an increasing fraction of loci switching to the nucleated OFF state asynchronously over time in the cold. The PRC2 accessory (VEL) proteins VIN3 and VRN5 contain head-to-tail polymerisation domains that are required for the switch to epigenetic silencing (Fiedler et al, 2022) (Fig. 2, top right). The polymerisation increases the avidity of PRC2 to nucleosomes for robust chromatin association through a molecular velcro mechanism (Payne-Dwyer et al, 2025; Schulten et al, 2025). We view the head-to-tail polymerisation mechanism as enhancing the feedback necessary for the epigenetic switch and subsequent memory, with the proteins themselves potentially retained through replication and therefore being directly part of the memory mechanism (Lövkvist et al, 2021). A similar protein-based bridging and oligomerisation mechanism has previously been proposed, but instead utilising components of the PRC1 system (Lo et al, 2012). Nevertheless, for FLC, this configuration is still ultimately unstable, leading to FLC reactivation over a period of ~1 week in the warm (Lövkvist et al, 2021; Yang et al, 2017), presumably due to the relatively low copy number of memory-storing elements in the spatially restricted nucleation region, which can be easily lost during replicative dilution.

The metastable state at FLC is then converted into a long-term epigenetically silenced OFF-state memory through the spreading of H3K27me3 across the FLC gene body, which requires an active cell cycle as the plants grow following return to warmer temperatures (Fig. 2, bottom right). At this stage, the histone-modifying enzymes may loop out from the nucleation site to locally distribute their histone marks through read-write feedback both during spreading and after each replicative dilution event. This spreading locks in a higher copy number of memory elements, thus ensuring strong memory stability in noisy conditions and through many DNA replications when memory elements are diluted. The spreading process is again digital and in cis, as judged by single-cell fluorescence imaging (Berry et al, 2015). After cell division has ceased, the system is thought to transition to a perpetuated state where H3K27me3 is sufficient to hold digital OFF-state memory of silencing without any other extra memory elements (Qüesta et al, 2020). Note that FLC copies that have not been epigenetically targeted by H3K27me3 nucleation/spreading, reactivate expression in warm conditions after cold. Furthermore, as seeds subsequently develop in warm conditions, the epigenetic switch is reversed, and FLC expression also resumes, in a process involving an H3K27me3 demethylase, ensuring each generation of plants requires a prolonged period of cold to flower (Crevillén et al, 2014).

A nucleation and spreading mechanism is also thought to occur during the developmental transition discussed above, as judged from spatially-resolved H3K27me3 time courses (Menon et al, 2024). However, for FLC, the proteins that mediate this switch are currently unknown and do not include VIN3, which is only expressed during prolonged cold. This mechanism is also considered to be the basis of initiation and maintenance (replication-dependent phase) of Polycomb silencing in mammals (Tatarakis et al, 2025; Oksuz et al, 2018; Veronezi and Ramachandran, 2024).

## Summary and outlook

In this review, we have discussed the fundamental distinction between graded and ON/OFF control for quantitative gene expression, with the ON/OFF modality important for trans-factor and histone modification-based epigenetic memory systems. As we have seen, both graded and ON/OFF modalities are perfectly compatible with both bursty and non-bursty transcriptional control. Dissecting the regulation of Arabidopsis FLC has been particularly insightful, as it exhibits both graded and ON/OFF modalities with a variably timed switch to an epigenetically silenced state.

We firmly believe that a deeper quantitative understanding of gene expression and epigenetic memory necessarily requires a more widespread consideration of ON/OFF versus graded control. Without fully incorporating this fundamental distinction, it will not be possible to properly unravel the circuits regulating these fundamental processes. To make further progress, understanding how graded versus ON/OFF regulation can be achieved in terms of the detailed individual steps of transcription is now an important goal.

Central to this effort will be increased use of interdisciplinary methodologies, including theoretical modelling. In the presence of multiple interacting feedback interactions, the system behaviours become hard to intuit. For example, for histone modification-based epigenetic memory states, do the known feedbacks permit

persistent memory states, or are additional components/interactions necessary? How long can memory states last before fluctuations compromise their fidelity? Modelling can be a powerful tool to cut through the complexity to unravel how the various feedbacks connect to the system output, yielding quantitative, experimentally testable predictions. These questions are often approached through simulations that incorporate the high levels of stochasticity inherent in molecular systems with low copy numbers (Menon and Howard, 2022), though on occasion, analytic calculations are also possible (Movilla Miangolarra and Howard, 2024). Inevitably, model construction involves simplification of the underlying biology: there is little point in constructing models that simply mirror the extreme complexity of the biology they are attempting to rationalise. However, over-simplification is also a risk, attempting to load too much functionality onto too limited an array of components. The most successful models balance these conflicting demands and open new predictive perspectives on mechanism.

We expect that continued investigation of the *FLC* locus will yield further insights into, for example, epigenetic memory mechanisms and components. As well as providing fundamental information on gene regulation, such progress will be extremely useful for gene editing/designing new *FLC* alleles in a range of crops adapted to future climates (Shindo et al, 2005, 2006; Hepworth et al, 2020). However, similarly in-depth studies of other example loci are urgently needed, using the distinction between the graded versus ON/OFF modalities as a necessary overarching paradigm. Understanding how multiple co-transcriptional mechanisms are integrated at individual loci and at a cellular resolution will be hugely important, as differential contributions at different genes and environmental contexts will otherwise mean key messages are lost when averaging over the whole genome. An improved understanding of transcription and epigenetic memory can then be put to work in synthetic biology contexts, utilising tools that are increasingly available for precision genome engineering in both analogue and digital contexts (Noviello et al, 2023; Bintu et al, 2016; Palacios et al, 2025; Policarpi et al, 2024).

## Peer review information

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

## Acknowledgements

We would like to thank Rob Klose, Eduardo Mateo-Bonmati, Govind Menon, Miguel Montez-Mariano-Coelho and Anna Schulten for critical reading of the manuscript. CD acknowledges funding from European Research Council Advanced Grant (EPISWITCH, 833254), Wellcome Trust (210654/Z/18/Z), and a Royal Society Professorship (RP\R1\180002). MH acknowledges funding from the Wellcome Trust (302160/Z/23/Z) and from the Wellcome Human Developmental Biology Initiative. CD and MH are both funded by the BBSRC Institute Strategic Programme BRiC (BB/P013511/1).

## Author contributions

**Caroline Dean**: Conceptualisation; Funding acquisition; Writing—original draft; Writing—review and editing. **Martin Howard**: Conceptualisation; Funding acquisition; Writing—original draft; Writing—review and editing.

## Disclosure and competing interests statement

The authors declare no competing interests.

