## [Peer Review File · The EMBO Journal]

Graded versus ON/OFF control in quantitative gene expression and epigenetic memory

Caroline Dean and Martin Howard

Corresponding author(s): Martin Howard (martin.howard@jic.ac.uk) , Caroline Dean (caroline.dean@jic.ac.uk)

Review Timeline:

Submission Date:	21st Oct 25
Editorial Decision:	21st Nov 25
Revision Received:	15th Jan 26
Editorial Decision:	12th Feb 26
Revision Received:	12th Feb 26
Accepted:	25th Feb 26

Editor: William Teale

Transaction Report:

Dear Martin and Caroline,

Thank you again for the submission of your manuscript entitled "Graded versus ON/OFF control in quantitative gene expression and epigenetic memory" and for your patience during the review process. We have now received reports from two referees, which I copy below.

Both sets of comments contain some constructive suggestions and recommendations that identify places in which some more detail would be helpful. The issues highlighted will require your attention before your manuscript can be published in The EMBO Journal.

Based on the overall interest expressed in the reports, however, I would like to invite you to address the comments of all referees in a revised version of the manuscript. Please, follow the instructions below when preparing your manuscript for resubmission.

I will carefully proof-read your final version and arrange for both figures to be professionally re-drawn. If you would like to Zoom about the reports, I am always available - just drop me a line.

When preparing your letter of response to the referees' comments, please bear in mind that this will form part of the Review Process File, and will therefore be available online to the community. For more details on our Transparent Editorial Process, please visit our website: <https://www.embopress.org/page/journal/14602075/authorguide#transparentprocess>. Again, please contact me at any time during revision if you need any help or have further questions.

Thank you very much again for the opportunity to consider your work for publication. I look forward to your revision.

Best regards,

William

William Teale, Ph.D.
Editor
The EMBO Journal

When submitting your revised manuscript, please carefully review the instructions below and include the following items:

- 1) a .docx formatted version of the manuscript text (including legends for main figures, EV figures and tables). Please make sure that the changes are highlighted to be clearly visible.
- 2) individual production quality figure files as .eps, .tif, .jpg (one file per figure).
- 3) a .docx formatted letter INCLUDING the reviewers' reports and your detailed point-by-point response to their comments. As part of the EMBO Press transparent editorial process, the point-by-point response is part of the Review Process File (RPF), which will be published alongside your paper.
- 4) a complete author checklist, which you can download from our author guidelines ([https://wol-prod-cdn.literatumonline.com/pb-assets/embo-site/Author Checklist%20-%20EMBO%20J-1561436015657.xlsx](https://wol-prod-cdn.literatumonline.com/pb-assets/embo-site/Author%20Checklist%20-%20EMBO%20J-1561436015657.xlsx)). Please insert information in the checklist that is also reflected in the manuscript. The completed author checklist will also be part of the RPF.
- 5) Please note that all corresponding authors are required to supply an ORCID ID for their name upon submission of a revised manuscript.
- 6) Our journal encourages inclusion of *data citations in the reference list* to directly cite datasets that were re-used and obtained from public databases. Data citations in the article text are distinct from normal bibliographical citations and should directly link to the database records from which the data can be accessed. In the main text, data citations are formatted as follows: "Data ref: Smith et al, 2001" or "Data ref: NCBI Sequence Read Archive PRJNA342805, 2017". In the Reference list, data citations must be labeled with "[DATASET]". A data reference must provide the database name, accession number/identifiers and a resolvable link to the landing page from which the data can be accessed at the end of the reference. Further instructions are available at .

Further general instructions for preparing your revised manuscript:

Read our guidance for manuscript revisions and related editorial policies: <https://link.springer.com/journal/44318/submission-guidelines#cms-Revised-submissions>

<https://media.springernature.com/original/springer-cms/rest/v1/content/27825798/data/v1>

- a point-by-point response to the referees' comments, with a detailed description of the changes made (as a word file).
- a word file of the manuscript text.
- individual production quality figure files (one file per figure)
- a complete author checklist
- Expanded View files (replacing Supplementary Information)
- a Reagents and Tools Table as part of the Methods section

We realize that it is difficult to revise to a specific deadline. In the interest of protecting the conceptual advance provided by the work, we recommend a revision within 3 months (19th Feb 2026). Please discuss the revision progress ahead of this time with the editor if you require more time to complete the revisions. Use the link below to submit your revision:

Referee #1:

The manuscript entitled "Graded versus ON/OFF control in quantitative gene expression and epigenetic memory" by Dean and Howard introduce a new emphasis of classification into GRADED vs ON/OFF control of quantitative gene expression that is viewed to be independent of transcriptional bursts. This leads to authors to highlight the importance of ON/OFF control for trans-factor and histone modification-mediated epigenetic memory. Although, the idea is timely as the molecular basis of the transcription process has sufficiently advanced, the current text falls short in clarifying what the new classification will facilitate in the future as opposed to what has been possible without it. To clarify I would broadly agree with the authors that this is a very productive area for computational modelling but some detail on how this modelling will help to understand questions in biology, ecology or other could be spelled out directly.

I can suggest a few points that the authors might consider for further polishing their work and helping readers appreciate key ideas.

1. I enjoyed reading the article, which was informative and had a good balance of conceptual discussion along sufficient biological detail. The text ends fairly abruptly after a tour de force of detail of molecular details and a concise summary, which left me wondering of how the concepts and insights would impact future work. I would suggest to add a brief future outlook at the end to capture the readers imagination and position the review in the scientific process.

2. The text covers a comprehensive and impressive number of ideas. Overall a insightful and interesting contribution. Possibly, the structure of the presentation could be improved for making it easier for the reader to follow key ideas through some sections of the discussion. One example is the sentence in the beginning "From the perspective of quantitative gene expression, the analogue modality is conceptually straightforward: each copy of a gene can express at a particular level and hence there is no fundamental difficulty in either an individual cell, or a collection of cells, generating specific levels of expression" which could be read in two ways." One way would mean the cell does not care how much or little a gene is expressed and will be fine. The other would be that the authors consider it trivial to maintain a level of a gene at a certain magnitude. The former case would make the

sentence pointless in the context of the discussion. However, the latter case would require an explanation as the biologically minded readership might find it surprising in the light of a highly complex regulatory system keeping gene expression levels comparable between cells of specific tissues between individuals. I suspect by defining the "difficulty" of what would make the reader properly instructed to take the meaning the authors intended.

3. One aspect of the article is the suggestion to use abstract concepts and computational modelling for deeper understanding biologic mechanisms. Yet, details of the computational methods are very limited in the text. This could be well justified by requirement to compose a concise treatment. However, if an opportunity would exist a panel could render how to think about how many factors a computational strategy would consider and the relevance of a choice of factors could be derived from contemporary understanding of molecular mechanisms. Transparently discussing opportunities and possibly limitations of the computational approach for directing future thinking and adoption of ideas could improve positioning of the review against earlier work.

Minor points

- a) It should be stated that FLC itself encodes a DNA binding factor that acts on a number of flowering genes. It would be interesting to know if it also regulates its own expression or if there are FLC target genes that are not connected to flowering.
- b) Depending on the intended readership bursty and non-bursty modes of transcription could briefly be explained as it seems to be prominently used in an exclusion clause to define the modalities of gene expression considered in the article.
- c) The sentence on page 5 "ON/OFF systems, due to the reduced space of possible outcomes, demonstrate enhanced robustness regardless of their molecular implementation: " could be rephrased as state-space is not considered elsewhere in the paragraph and the reasoning is incomplete in a biologic context.
- d) The text on page 9 "activating modifications such as H3K36me3" could be misleading. Although K36me3 is associated with active genes it is thought to act repressive to prevent transcription initiation within the transcription unit. In mammalian cell cultures recruitment of DNMT3 DNA methyltransferase by a K36me3 binding domain has been proposed
- e) Page 15 discusses the regulation of H3K27 methylation and relation to gene silencing. An additional prominent observation could be added here as the loss of K27me3 is also seen on Xi when the Xist gene is removed (Ref 16 in doi: 10.1126/science.1084274) showing that there is little stability - likely only one extra cell division for K27me3 - e.g. Figure 5 in doi: 10.1371/journal.pbio.0020171 - For the argumentation the Xi might be worth considering as it has a very extensive region of K27 methylation. Since the literature seems to somewhat constrain polymerization models of PcG proteins of memory - there might be an opportunity to dissect the memory mechanism in Arabidopsis FLC regulation that could be pointed to in a future outlook section.

Referee #2:

The authors propose an interesting hypothesis regarding the difference between a ON/OFF model of transcription versus a graded analog model. The second part of the manuscript is dedicated to illustrating the model using their own work on FLC downregulation in Arabidopsis. While this second part provides an excellent summary of the past work, it does not connect so well with the first section discussing models of gene expression - not silencing. These are not necessarily mirror-image of each other and this point causes a chasm with the first part.

The initial section discusses models of active transcription comparing previously proposed views on the integration of bursts of transcription for example with the opposition between the ON/OFF versus the analog model. The broad statements of the first part are not always well supported by data as detailed below or the support comes several paragraphs too late. It would be better to restructure the work starting with the FLC section and then construct a model based on concrete examples taken from other studies rather than the imagination of the authors. Currently as the paper stands the initial part contains too many sweeping inaccurate statement as outlined below to be left in its current state.

"However, for the ON/OFF mode, clearly any individual diploid cell will be limited to only three levels of expression (OFF/OFF, ON/OFF, ON/ON), if each copy of the gene can be independently regulated, or only two (OFF/OFF, ON/ON), if both copies share the same regulation." And later "ON/OFF control is conceptually more complex and suffers from the problem that at the level of an individual gene copy or cell, only a few discrete levels of expression are possible (see above). Smoothly varying quantitative control can only be recovered at the level of collections of cells." - This is a very reductionist view of what could happen since cell could be in G1 with two copies or G2 with four copies and in the case of plant cells, due to endoreduplication the number of copies can vary up to several hundred. Since the authors works refer to plants, even the ON/OFF model could produce a smooth gradation of the transcripts depending in the number of copies. In addition the copies of cis-elements have the capacity to titrate out transcriptional regulators so one can imagine a much more complex view of the output of the ON/OFF

model than presented by the authors.

"We emphasise that both bursty and non-bursty output is completely compatible with our distinction into graded versus ON/OFF control for expression." Here the authors mix two concepts: the temporal control of transcription which overall achieves the final expression in a cell (integrations of the bursts) with their model which already takes this into account so indeed there is no reason these would be incompatible because the time scale of the models are totally different.

"the analogue versus ON/OFF distinction is perhaps an even more fundamental classification for gene regulation (than burst controlled transcription)." This statement is not really justified because the burst addresses a real phenomenon directly linked to mechanism whereas the model proposed by the authors relates to the global transcription level at the level of tissue integrated over a population of cells. So the two types of regulations can not be compared and even less rated.

"Given the fundamental nature of ON/OFF versus graded regulation, it is perhaps surprising that this concept has not been discussed more prominently in the literature." These models have already discussed these at length in their previous publications PMID: 37466633, PMID: 33662809, PMID: 25955967.

"Given the fundamental nature of ON/OFF versus graded regulation, it is perhaps surprising that this concept has not been discussed more prominently in the literature. One reason is that elucidating the precise mode of regulation requires single-cell analysis: while this is feasible for mRNA/protein/DNA methylation measurements, it is impractical for histone modifications, at least with current technology. Measurements with bulk RNA-seq and ChIP-seq, which are still by far the most prevalent, cannot distinguish these modes." Here the authors imply that histone modifications would be causal whereas there is a consensus from the literature that histone modifications associated with active transcription result from the ongoing event of transcription and are thus a reflection but not a cause -although indeed as the authors state later they can autoregulate transcriptional elongation.

"ON/OFF systems, due to the reduced space of possible outcomes, demonstrate enhanced robustness regardless of their molecular implementation: in principle only very large fluctuations will be able to flip the state of the system from ON to OFF or vice-versa (Figure 1B, top). Graded, analogue systems, on the other hand, will display far less robustness (Figure 1B, bottom)." - Could the authors first state data supporting this view?

"This greater responsiveness may be advantageous when the system needs to respond to inputs. However, one scenario where such sensitivity will be problematic is in systems that must exhibit memory, i.e., in epigenetic regulation. Here, by definition, a gene must remember its expression level even when the originating signals that first set that level have disappeared." This is an idea already proposed by Mark Ptashne in his review of the topic PMID: 24385432. This is cited in later paragraphs but should be evoked much earlier. Epigenetic regulation can be triggered by an autoactivating loop that does not necessarily require chromatin-based regulation.

"However, as also discussed below, memory based on DNA methylation may escape this necessity, due to the high fidelity and very low fluctuations present in the maintenance dynamics of DNA methylation." And later "For DNA CG methylation, maintenance occurs at the level of each individual CG site, where hemi-methylated CGs are recognised after replication and remethylated. This is in contrast to the histone modification-based systems, where it is only the collective state of the histone modifications that stores memory rather than any individual histone modification, which might be stochastically lost at DNA replication." - The authors have a very simplified view of the fidelity of the copy of DNA methylation that is now known to be fully restored not immediately after DNA replication as they imply but much later late G2 at best (see PMID: 32581343 for animal cells). By contrast, histone modifications are inherited (half levels) at the DNA replication fork (Anja Groth's work) and thus at least as efficiently transmitted as DNA methylation from mother to daughter cells. The remethylation of each hemimethylated CpG site does not necessarily happen an overall remethylating DNA and redepositing histone modifications are very similar in essence and this does not justify the last paragraph.

"During this phase of the cell cycle, approximately half of the histone modifications will be lost due to the distribution of parental nucleosomes between the two daughter DNA strands." And "In fact, the requirement may be even more complex, as a halving of histone modification levels is believed to only occur on average (Annunziato, 2005). Due to the stochastic inheritance of nucleosomes, the fraction of modifications surviving on a given daughter strand may be either greater or smaller than a half (Annunziato, 2005). An analogue histone modification memory system would have to register this survival information and then incorporate it into the post-replication recovery in order to reacquire the appropriate, pre-replication graded level of histone modifications". Such statements are not supported by more recent data from the Groth and the Almouzni labs recording histone and histone modification dynamics at the DNA replication fork showing the histone modifications are restored rapidly after DNA replication and before the next division.

"Key to maintenance of the OFF state is the read-write action of PRC2, where the complex can "read" existing H3K27me3 marks, and then undergo allosteric activation to "write" more of the mark nearby in a positive feedback loop. Similar feedback loops also exist for H2Aub via PRC1, as well as crosstalk between these marks." This is a huge simplified and distorted view of an extremely complex and controversial series of works in animals and plants showing that depending on the nature of the PRC1 and PRC2 complexes, cross talks may or may not exist. Please cite the relevant literature and tone down.

Dear William,

Thank you for forwarding the reviewer comments to us. We have now comprehensively rewritten the review to address the very constructive points raised by both reviewers. We think this has substantially raised the quality of the review. A detailed rebuttal can be found below. We would also very much appreciate it if the two figures could be professionally re-drawn!

Best wishes,

Caroline and Martin.

Referee #1:

Although, the idea is timely as the molecular basis of the transcription process has sufficiently advanced, the current text falls short in clarifying what the new classification will facilitate in the future as opposed to what has been possible without it. To clarify I would broadly agree with the authors that this is a very productive area for computational modelling but some detail on how this modelling will help to understand questions in biology, ecology or other could be spelled out directly.

We have now expanded the "Summary and Outlook" section to discuss computational modelling, highlighting the role it can play to dissect complex mechanisms in biology.

1. I enjoyed reading the article, which was informative and had a good balance of conceptual discussion along sufficient biological detail. The text ends fairly abruptly after a tour de force of detail of molecular details and a concise summary, which left me wondering of how the concepts and insights would impact future work. I would suggest to add a brief future outlook at the end to capture the readers imagination and position the review in the scientific process.

Thank you – this is a good suggestion: we have reworked the final section, retitling it as "Summary and Outlook", adding some further forward-looking discussion outlining how our insights may impact future work.

2. The text covers a comprehensive and impressive number of ideas. Overall a insightful and interesting contribution. Possibly, the structure of the presentation could be improved for making it easier for the reader to follow key ideas through some sections of the discussion. One example is the sentence in the beginning "From the perspective of quantitative gene expression, the analogue modality is conceptually straightforward: each copy of a gene can express at a particular level and hence there is no fundamental difficulty in either an individual cell, or a collection of cells, generating specific levels of expression" which could be read in two ways." One way would mean the cell does not care how much or little a gene is expressed and will be fine. The other would be that the authors consider it trivial to maintain a level of a gene at a certain magnitude. The former case would make the sentence pointless in the context of the discussion. However, the latter case would require an explanation as the biologically minded readership might find it surprising in the light of a highly complex regulatory system keeping gene expression levels comparable between cells of specific tissues between individuals. I suspect by defining the "difficulty" of what would make the reader properly instructed to take the meaning the authors intended.

The sentence referred to by the reviewer was indeed poorly worded. Of course, maintaining a particular level of expression consistently between cells and individuals can be difficult to

achieve and is most certainly important for cells, as we now clearly state. What we meant to say is that because each gene copy is assumed to be capable of having analogue, graded expression, having tuneable expression levels at a tissue level is also possible in principle without the strict need for any further regulatory concepts. For the digital case, on the other hand, the fraction of cells expressing must be regulated. We have rewritten this section to make this clearer.

3. One aspect of the article is the suggestion to use abstract concepts and computational modelling for deeper understanding biologic mechanisms. Yet, details of the computational methods are very limited in the text. This could be well justified by requirement to compose a concise treatment. However, if an opportunity would exist a panel could render how to think about how many factors a computational strategy would consider and the relevance of a choice of factors could be derived from contemporary understanding of molecular mechanisms. Transparently discussing opportunities and possibly limitations of the computational approach for directing future thinking and adoption of ideas could improve positioning of the review against earlier work.

We did indeed try to keep details of computational modelling to a relatively low level, to maintain simplicity as well as to keep things concise. However, we agree that some additional discussion would be useful. We have therefore added a new paragraph on the potential of computational modelling, as well as its limitations, into the main text in the "Summary and Outlook".

Minor points

a) It should be stated that FLC itself encodes a DNA binding factor that acts on a number of flowering genes. It would be interesting to know if it also regulates its own expression or if there are FLC target genes that are not connected to flowering.

We have added extra text to the *FLC* section to discuss these issues. FLC is not thought to regulate its own expression. As we now mention, there are many FLC targets that are unrelated to flowering. One study (Deng et al, PNAS 2011) found 505 FLC binding sites genome-wide, with genes identified as FLC targets broadly involved in developmental pathways throughout the life history of the plant. For example, FLC was found to be involved in vegetative development (via SPL15), delaying juvenile to adult progression.

b) Depending on the intended readership bursty and non-bursty modes of transcription could briefly be explained as it seems to be prominently used in an exclusion clause to define the modalities of gene expression considered in the article.

We explain the meaning of bursty and non-bursty transcription in 4th paragraph of the "Graded versus ON/OFF modes of quantitative gene expression control" section. As suggested, however, we have now added a little more detail to this discussion.

c) The sentence on page 5 "ON/OFF systems, due to the reduced space of possible outcomes, demonstrate enhanced robustness regardless of their molecular implementation: " could be rephrased as state-space is not considered elsewhere in the paragraph and the reasoning is incomplete in a biologic context.

We have rewritten this passage, deleting the state-space discussion, to improve our reasoning (see also our reply to Referee #2 below).

d) The text on page 9 "activating modifications such as H3K36me3" could be misleading. Although K36me3 is associated with active genes it is thought to act repressive to prevent

transcription initiation within the transcription unit. In mammalian cell cultures recruitment of DNMT3 DNA methyltransferase by a K36me3 binding domain has been proposed

Thank you: due to the more nuanced role of H3K36me, we have deleted it from this section.

e) Page 15 discusses the regulation of H3K27 methylation and relation to gene silencing. An additional prominent observation could be added here as the loss of K27me3 is also seen on Xi when the Xist gene is removed (Ref 16 in doi: 10.1126/science.1084274) showing that there is little stability - likely only one extra cell division for K27me3 - e.g. Figure 5 in doi: 10.1371/journal.pbio.0020171 - For the argumentation the Xi might be worth considering as it has a very extensive region of K27 methylation. Since the literature seems to somewhat constrain polymerization models of PcG proteins of memory - there might be an opportunity to dissect the memory mechanism in Arabidopsis FLC regulation that could be pointed to in a future outlook section.

Thank you for pointing out the relevance of Xi. We have worked a discussion of H3K27me at Xi into a discussion of timescales in the "ON/OFF control in the Polycomb memory system" section. We have also highlighted that *FLC* is a very attractive system in which to dissect memory mechanisms (included in the "Summary and Outlook").

Referee #2:

While this second part provides an excellent summary of the past work, it does not connect so well with the first section discussing models of gene expression - not silencing. These are not necessarily mirror-image of each-other and this point causes a chasm with the first part.

We have now improved the connection between the first and second parts of the review. The first part discusses two different modalities of quantitative gene expression and how these modalities apply to systems exhibiting epigenetic memory, including memory of silencing. As *FLC* exhibits both modalities, with a switch between them, it is an excellent illustration of the concepts from the first part.

The initial section discusses models of active transcription comparing previously proposed views on the integration of bursts of transcription for example with the opposition between the ON/OFF versus the analog model. The broad statements of the first part are not always well supported by data as detailed below or the support comes several paragraphs too late. It would be better to restructure the work starting with the *FLC* section and then construct a model based on concrete examples taken from other studies rather than the imagination of the authors. Currently as the paper stands the initial part contains too many sweeping inaccurate statement as outlined below to be left in its current state.

We understand the referee's point of view that the *FLC* section should come first. However, if we were to reverse the order, the review would look, at least at first sight, as being just another review on *FLC* regulation. Our purpose is indeed to review the mechanistic basis of *FLC* expression and memory but within the conceptual framework of analogue versus digital expression. Without first spelling out what we mean by analogue and digital control, we think it is hard to make sense of how *FLC* is regulated. Nevertheless, as discussed above, we agree with the reviewer that the connections between the two sections were not as clearly articulated as they could have been and we have endeavoured to improve this connection in the revision. We have also addressed and increased the precision of some of statements that were inaccurate (see below).

"However, for the ON/OFF mode, clearly any individual diploid cell will be limited to only three levels of expression (OFF/OFF, ON/OFF, ON/ON), if each copy of the gene can be

independently regulated, or only two (OFF/OFF, ON/ON), if both copies share the same regulation." And later "ON/OFF control is conceptually more complex and suffers from the problem that at the level of an individual gene copy or cell, only a few discrete levels of expression are possible (see above). Smoothly varying quantitative control can only be recovered at the level of collections of cells." - This is a very reductionist view of what could happen since cell could be in G1 with two copies or G2 with four copies and in the case of plant cells, due to endoreduplication the number of copies can vary up to several hundred. Since the authors works refer to plants, even the ON/OFF model could produce a smooth gradation of the transcripts depending in the number of copies. In addition the copies of cis-elements have the capacity to titrate out transcriptional regulators so one can imagine a much more complex view of the output of the ON/OFF model than presented y the authors.

The referee raises some very interesting points here. Firstly, with respect to the G1 (2 copies) versus G2 (4 copies). If switching has occurred in a previous cell cycle or within G1, during G2 there will still only be three states (ONx4, ONx2/OFFx2 or OFFx4). If switching occurs during G2, it is true that there can be a more complex picture with, for example, only 1 out of 4 copies ON. However, this is only a transient, as at the next cell cycle the system will resolve into two of the three standard states (ON/OFF and OFF/OFF). Secondly, the referee raises the issue of endoreduplication. It is of course correct that in plants there may be much high gene copy numbers. In that case, the digital regulation could indeed give effective analogue control. Titration of transcriptional regulators could add a further level of complexity. We now discuss these issues briefly in the "Graded versus ON/OFF modes of quantitative gene expression control" section.

"We emphasise that both bursty and non-bursty output is completely compatible with our distinction into graded versus ON/OFF control for expression." Here the authors mix two concepts: the temporal control of transcription which overall achieves the final expression in a cell (integrations of the bursts) with their model which already takes this into account so indeed there is no reason these would incomptabilble because the time scale of the models are totally different.

We completely agree! Nevertheless, when we have presented our conceptual picture to colleagues in talks and discussions, this issue has arisen multiple times, so we think it is worth discussing.

"the analogue versus ON/OFF distinction is perhaps an even more fundamental classification for gene regulation (than burst controlled transcription)." This statement is not really justified because the burst addresses a real phenomenon directly linked to mechanism whereas the model proposed by the authors relates to the global transcription level at the level of tissue integrated over a population of cells. So the two types of regulations can not be compared and even less rated.

We have dropped the claim that analogue versus ON/OFF control is more fundamental than bursty/non-bursty, which was perhaps an overly-combative opinion, while retaining the discussion of compatibility (see above).

"Given the fundamental nature of ON/OFF versus graded regulation, it is perhaps surprising that this concept has not been discussed more prominently in the literature." These model have already discussed these at length in their previous publications PMID: 37466633, PMID: 33662809, PMID: 25955967.

We agree that we have discussed some of these issues before in previous publications, as we now state in the text, but not in the wide-ranging and comprehensive way we strive for

here. Furthermore, these ideas have not yet percolated into the wider community. This is the fundamental reason why we have written this review.

"Given the fundamental nature of ON/OFF versus graded regulation, it is perhaps surprising that this concept has not been discussed more prominently in the literature. One reason is that elucidating the precise mode of regulation requires single-cell analysis: while this is feasible for mRNA/protein/DNA methylation measurements, it is impractical for histone modifications, at least with current technology. Measurements with bulk RNA-seq and ChIP-seq, which are still by far the most prevalent, cannot distinguish these modes." Here the authors imply that histone modifications would be causal whereas there is a consensus from the literature that histone modifications associated with active transcription results from the ongoing event of transcription and are thus a reflections but not a cause -although indeed as the authors state later they can autoregulate transcriptional elongation.

We have now deleted this paragraph as it was somewhat less important, and we wanted to make room for other edits.

"ON/OFF systems, due to the reduced space of possible outcomes, demonstrate enhanced robustness regardless of their molecular implementation: in principle only very large fluctuations will be able to flip the state of the system from ON to OFF or vice-versa (Figure 1B, top). Graded, analogue systems, on the other hand, will display far less robustness (Figure 1B, bottom)." - Could the authors first state data supporting this view?

We now emphasise in the text that this is a very general conceptual argument and is not based on any specific experimental data. We have also added some additional text to this argument to make it clearer.

"This greater responsiveness may be advantageous when the system needs to respond to inputs. However, one scenario where such sensitivity will be problematic is in systems that must exhibit memory, i.e., in epigenetic regulation. Here, by definition, a gene must remember its expression level even when the originating signals that first set that level have disappeared." This is an idea already proposed by Mark Ptashne in his review of the topic PMID: 24385432 . This is cited in later paragraphs but should be evoked much earlier. Epigenetic regulation can be triggered by an autoactivating loop that does not necessary require chromatin-based regulation.

We certainly agree that epigenetic regulation does not necessarily require chromatin-based control and that fact is prominently discussed in the text. We have also moved up the Ptashne citation as requested.

"However, as also discussed below, memory based on DNA methylation may escape this necessity, due to the high fidelity and very low fluctuations present in the maintenance dynamics of DNA methylation." And later "For DNA CG methylation, maintenance occurs at the level of each individual CG site, where hemi-methylated CGs are recognised after replication and remethylated. This is in contrast to the histone modification-based systems, where it is only the collective state of the histone modifications that stores memory rather than any individual histone modification, which might be stochastically lost at DNA replication." - The authors have a very simplified view of the fidelity of the copy of DNA methylation that is now known to be fully restored not immediately after DNA replication as they imply but much later late G2 at best (see PMID: 32581343 for animals cells) . By contrast, histone modifications are inherited (half levels) at the DNA replication fork (Anja Groth's work) and thus at least as efficiently transmitted as DNA methylation from mother to daughter cells. The remethylation of each hemimethylated CpG site does not necessary

happen an overall remethylating DNA and redepositing histone modifications are very similar in essence and this does not justify the last paragraph.

The referee makes some good points here. We did not mean to claim that hemi-methylated DNA is immediately remethylated after replication. Indeed, this process may take longer in some systems. We have clarified the text to make this aspect clearer. However, as long as re-methylation of individual hemi-methylated sites reliably happens before the next replication event then individual mCG sites are reliably inherited. This is the case for trans-generationally inherited mCG in Arabidopsis and in this case, for example, analogue memory is certainly possible (Briffa et al, Cell Systems 2023). However, in other cases, where re-methylation of hemi-methylated DNA is unreliable, the inheritance of DNA methylation memory states will indeed be much more similar to the histone modification case. In these cases, strong positive feedback will be needed to stabilise the memory states against methylation loss (like the read-write feedback for histone modification memory). As we mention in the text, this may be the case for human mCG. We have rewritten the text of this section to address these important points.

"During this phase of the cell cycle, approximately half of the histone modifications will be lost due to the distribution of parental nucleosomes between the two daughter DNA strands." And "In fact, the requirement may be even more complex, as a halving of histone modification levels is believed to only occur on average (Annunziato, 2005). Due to the stochastic inheritance of nucleosomes, the fraction of modifications surviving on a given daughter strand may be either greater or smaller than a half (Annunziato, 2005). An analogue histone modification memory system would have to register this survival information and then incorporate it into the post-replication recovery in order to reacquire the appropriate, pre-replication graded level of histone modifications". Such statements are not supported by more recent data from the Groth and the Almouzni labs recording histone and histone modification dynamics at the DNA replication fork showing the histone modifications are restored rapidly after DNA replication and before the next division.

The argument we are outlining here concerns the difficulties an analogue memory system would have in dealing with the dilution of histone marks at DNA replication. Of course, as pointed out by the reviewer, the actual histone modification dynamics are driven by strong positive read/write feedback which generates digital ON/OFF memory with rapid refilling of the lost histone marks after replication (although the dynamics are believed to be somewhat slower for H3K27me3) (Reverón-Gómez et al. 2018). We have edited the text here to make it clear that we are discussing the hypothetical possibility of analogue histone modification memory, why such a system is problematic, and therefore why the overall system is digital with strong read-write feedback, as seen experimentally.

"Key to maintenance of the OFF state is the read-write action of PRC2, where the complex can "read" existing H3K27me3 marks, and then undergo allosteric activation to "write" more of the mark nearby in a positive feedback loop. Similar feedback loops also exist for H2AUb via PRC1, as well as crosstalk between these marks." This is a huge simplified and distorted view of an extremely complex and controversial series of works in animals and plants showing that depending on the nature of the PRC1 and PRC2 complexes, cross talks may or may not exist. Please cite the relevant literature and tone down.

We agree that this view is somewhat simplified. However, we do not want to devote too much text to the fundamentals of Polycomb regulation, which have been well reviewed elsewhere (e.g., Uckelmann and Davidovich, 2021). We believe that the read/write feedback of H3K27me3/PRC2 and H2AUb/PRC1 are well established. Crosstalk between the two has also been extensively reported. However, we agree that H3K27me3 is not thought to feed

through to H2AUb levels, as H3K27me3 only appears to interact with cPRC1 which is ineffective in adding H2AUb, whereas H2AUb can feed through to H3K27me3 via PRC2 with the H2AUb-binding JARID2 subunit (Shi et al, 2024). We have added these references to the manuscript and amended the text appropriately to tone down our comments.

Dear Martin,

Many thanks for submitting a revised version of your manuscript. I have now had a chance carefully to go through this in light of both referees' reports and your responses to them and can see that you have addressed all concerns satisfactorily. Before I can finally accept the manuscript though, it would be very helpful if you would reorganize the reference list alphabetically according to our guidelines for authors (<https://link.springer.com/journal/44318/submission-guidelines#cms-Reference-guidelines>). If listing ten authors + et al. continues to cause problems, please let me know. I can then try to add the missing four names for the longer author lists from my side.

EMBO Press is an editorially independent publishing platform for the development of EMBO scientific publications.

Best wishes,

William

William Teale, PhD
Editor
The EMBO Journal
w.teale@embojournal.org

Read our guidance for manuscript revisions and related editorial policies: <https://link.springer.com/journal/44318/submission-guidelines#cms-Revised-submissions>

<https://media.springernature.com/original/springer-cms/rest/v1/content/27825798/data/v1>

- a point-by-point response to the referees' comments, with a detailed description of the changes made (as a word file).
- a word file of the manuscript text.
- individual production quality figure files (one file per figure)
- a complete author checklist
- Expanded View files (replacing Supplementary Information)
- a Reagents and Tools Table as part of the Methods section

Please remember: Digital image enhancement is acceptable practice, as long as it accurately represents the original data and conforms to community standards. If a figure has been subjected to significant electronic manipulation, this must be noted in the figure legend or in the 'Methods' section. The editors reserve the right to request original versions of figures and the original images that were used to assemble the figure.

We realize that it is difficult to revise to a specific deadline. In the interest of protecting the conceptual advance provided by the work, we recommend a revision within 3 months (13th May 2026). Please discuss the revision progress ahead of this time with the editor if you require more time to complete the revisions. Use the link below to submit your revision:

The authors have addressed all minor editorial requests.

Dear Martin and Caroline,

I am pleased to inform you that your manuscript has been accepted for publication in the EMBO Journal.

Many thanks for a really interesting piece!

You may qualify for financial assistance for your publication charges - either via a Springer Nature fully open access agreement or an EMBO initiative. Check your eligibility: <https://link.springer.com/journal/44318/how-to-publish-with-us>

Best wishes,

William

William Teale, PhD
Editor
The EMBO Journal
w.teale@embojournal.org

Please note that it is The EMBO Journal policy for the transcript of the editorial process (containing referee reports and your response letters) to be published as an online supplement to each paper. If you should prefer removal of any referee-only figures included in the point-by-point response(s), e.g. because they may still be used for future publication or because they have been reproduced from published work by others, please do let us know immediately via response email.
More information is available here: <https://link.springer.com/partners/embo-press/editorial-policies#Peer%20review>